# Insights into household transmission of SARS-CoV-2 from a population-based serological survey

Qifang Bi[1], Justin Lessler [1,24], Isabella Eckerle[2,3], Stephen A. Lauer[1], Laurent Kaiser[2,4,5], Nicolas Vuilleumier[5,6], Derek A. T. Cummings [7,8], Antoine Flahault[9,10,11], Dusan Petrovic[12,13,14], Idris Guessous[10,12], Silvia Stringhini [10,12,13], Andrew S. Azman [1,11,12,24✉], SEROCoV-POP Study Group*

Understanding the risk of infection from household- and community-exposures and the transmissibility of asymptomatic infections is critical to SARS-CoV-2 control. Limited previous evidence is based primarily on virologic testing, which disproportionately misses mild and asymptomatic infections. Serologic measures are more likely to capture all previously infected individuals. We apply household transmission models to data from a cross-sectional, household-based population serosurvey of 4,534 people ≥5 years from 2,267 households enrolled April-June 2020 in Geneva, Switzerland. We found that the risk of infection from exposure to a single infected household member aged ≥5 years (17.3%,13.7-21.7) was more than three-times that of extra-household exposures over the first pandemic wave (5.1%,4.5-5.8). Young children had a lower risk of infection from household members. Working-age adults had the highest extra-household infection risk. Seropositive asymptomatic household members had 69.4% lower odds (95%CrI,31.8-88.8%) of infecting another household member compared to those reporting symptoms, accounting for 14.5% (95%CrI, 7.2-22.7%) of all household infections.

[1] Department of Epidemiology, Johns Hopkins Bloomberg School of Public Health, Baltimore, MD, USA. [2] Geneva Center for Emerging Viral Diseases and Laboratory of Virology, Geneva University Hospitals, Geneva, Switzerland. [3] Department of Microbiology and Molecular Medicine, Faculty of Medicine, University of Geneva, Geneva, Switzerland. [4] Division of Infectious Diseases, Geneva University Hospitals, Geneva, Switzerland. [5] Department of Medicine, Faculty of Medicine, University of Geneva, Geneva, Switzerland. [6] Division of Laboratory Medicine, Geneva University Hospitals, Geneva, Switzerland. [7] Department of Biology, University of Florida, Gainesville, FL, USA. [8] Emerging Pathogens Institute, University of Florida, Gainesville, FL, USA. [9] Division of Tropical and Humanitarian Medicine, Geneva University Hospitals, Geneva, Switzerland. [10] Department of Health and Community Medicine, Faculty of Medicine, University of Geneva, Geneva, Switzerland. [11] Institute of Global Health, Faculty of Medicine, University of Geneva, Geneva, Switzerland. [12] Division of Primary Care Medicine, Geneva University Hospitals, Geneva, Switzerland. [13] University Centre for General Medicine and Public Health, University of Lausanne, Lausanne, Switzerland. [14] Centre for Environment and Health, School of Public Health, Department of Epidemiology and Biostatistics, Imperial College London, London, UK. [24]The authors contributed equally: Justin Lessler, Andrew S. Azman. *A list of authors and their affiliations appears at the end of the paper. ✉email: azman@jhu.edu

Household-centered studies provide an enumerable set of individuals known to be exposed to an infectious person, hence, they have played an important role for estimating key transmission properties of SARS-CoV-2. However, most published studies of SARS-CoV-2 household transmission rely on clinical disease (COVID-19), and/or PCR-based viral detection to identify infected individuals[1,2]. Due to the narrow time window after exposure in which RT-PCR can be highly sensitive[3], case ascertainment based on virologic testing may miss infections, especially those that are mild or asymptomatic[4]. This can lead to important biases and limit what can be studied, including underestimates of the importance of sub-clinical infections and household secondary attack rates[4].

Serologic studies provide an alternative tool for understanding SARS-CoV-2 transmission. Serological tests remain sensitive to detecting past infections well beyond the period when the virus is detectable[5–7], thereby providing a measure of whether individuals have ever been infected.

Virologic and serologic studies have each provided important insights into SARS-CoV-2 transmission. These include estimates of the household secondary attack rate (e.g., 17% in a meta-analysis[2]) and evidence of reduced infection rates among young children[2,8,9]. However, in general, these estimates do not distinguish between intra- and extra-household transmission nor do they provide an estimate of transmission risk from a single infected individual. A notable exception is a household study from Guangzhou, China[10], but this PCR-based study suffered from the limitations of virologic testing noted above. Hence, a number of critical gaps in the evidence remain, including the relative role of transmission between household members, the frequency of viral introductions into households from the community, the infectiousness of asymptomatic individuals, and the effect of age on transmission.

To help fill these gaps, we apply household transmission models to data from a cross-sectional, household-based population serosurvey of 4534 people from 2267 households in Geneva, Switzerland (SEROCoV-PoP). We provide a serology-based assessment of transmission between intra- and extra-household contacts, identify risk factors for infection and transmission and estimate the relative risk of asymptomatic transmission. By doing so, we provide important evidence for guiding the COVID-19 pandemic response.

## Results

Between April 3rd and June 30th, during the first wave of the SARS-CoV-2 pandemic in Geneva, 8344 individuals coming from 4393 households were successfully enrolled in the SEROCoV-POP study (Figs. 1 and S1)[11]. The median enrollment date was May 22nd, 86 days after the first case was detected in Geneva (February 26th, 2020). In 2267 of these households, all members of the household were eligible, available, and provided a blood sample for detection of anti-SARS-CoV-2 IgG antibodies by ELISA (4354 individuals). The majority of these households were either one (37.9%, n=860) or two (39.2%, n=889) person households (Fig. S2, Table S1). The median household size in our study (2.0, interquartile range [IQR]=1,2) was similar to the general population in Geneva canton (median=2.0, IQR=1,3)[12].

The median age of participants was 53 years (IQR=34,65), and 53.6% were female. Compared with the general canton population, our study sample included more individuals 50 years and older and fewer 20–49 year olds. Individuals in older age groups were more likely to live in smaller households: 94.6% (1100/1163) of people who were 65 years and older lived alone or in two-person households versus 44.5% (588/1302) of those 20–49 years old (Table 1). Our study sample, like that of the original

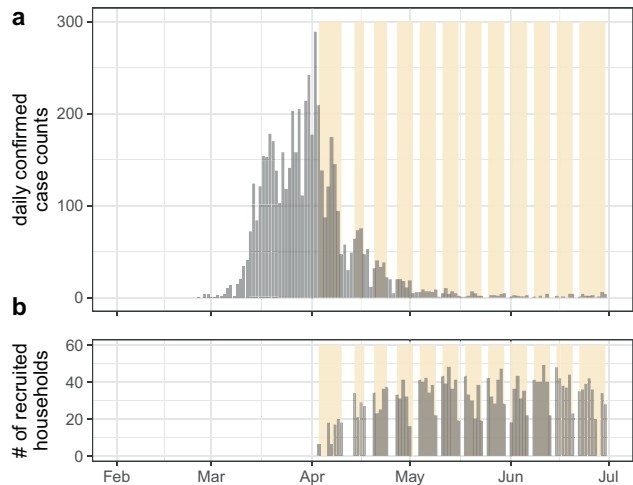

**Fig. 1 Epidemic curve and recruitment period of household serosurvey. a** daily confirmed COVID-19 cases reported in Geneva up to July 1st, 2020. **b** Daily number of recruited households over the 12-week study period. First detected case in Geneva canton was reported on February 26th, and the first epidemic wave lasted about two months. Yellow bands indicate time periods of study enrollment for each week. This includes all 4438 households enrolled in the SEROCoV-POP study, not restricted to the complete households used in these analyses for which serostatus of all household members were available.

SEROCoV-POP study, had a higher level of formal education than the general canton population with only 8.5% not having a high school degree or equivalent, compared with 23.5% in the general canton population (Table S5)[13].

Overall, 6.6% (298/4534) of individuals tested positive for SARS-CoV-2 anti-S1 IgG antibodies by ELISA. Of the 2267 households included in the analyses, 222 (9.8%) had at least one seropositive household member. The proportion of households with seropositive members increased from 4.8% (41/860) in households of size one, to 17.0% (39/229) in households of size three, and was relatively constant in larger households (Fig. S2, Table 1, Fig. S3). Symptoms consistent with COVID-19 were reported by 69.5% (207/298) of seropositive individuals although this was significantly lower in young children (37.5%, 3/8), similar to the results of an early modeling study[14].

We fit household transmission models and estimated that from the start of the epidemic in Geneva through the time of the serosurvey, the cumulative risk of infection from extra-household exposures was 5.1% (95% Credible Interval [CrI] 4.5–5.8%). The probability of being infected from a single infected household member was 17.3% (95% CrI 13.7–21.7%, Fig. 2).

The risk of being infected by a household member was the lowest among 5–9 years old and highest among those 65 years and older, with teenagers and working age adults sharing similar risks (Figs. 2, 3). Compared to 20–49 years olds, 5–9 years olds had less than half the odds of being infected by an infected household member (OR=0.4, 95%CrI 0.1–1.6), while those 65 years and older had nearly three times the odds (OR=2.7, 95%CrI 0.9–7.9). Though credible intervals on these estimates are wide, and both include the null value of 1, inclusion of age substantially improved model fit (ΔWAIC −14.8, Table S2). In contrast, the extra-household infection risk was the highest among working age adults (20–49 years olds). Compared to this group, 5–9 year olds (OR=0.5, 95%CrI 0.2–0.9) and those 65 years and older (OR=0.4, 95%CrI 0.3–0.6) had the lowest risk (Fig. 3, Tables S2 and S4). Models allowing for differential risk of transmission by

**Table 1 Number of recruited and seropositive individuals by age-group, sex and household size of the households they reside in.**

| | HH Size 1 sero +/N % (95% CI) | HH Size 2 sero +/N % (95% CI) | HH Size 3 sero +/N % (95% CI) | HH Size 4 sero +/N % (95% CI) | HH Size 5 +sero+/N % (95% CI) | Overall sero +/N % (95% CI) | Odds ratio for being seropositive |
|---|---|---|---|---|---|---|---|
| **HOUSEHOLDS** | | | | | | | |
| 0 seropositive | 819/860 95% (94–96) | 807/889 91% (89–93) | 190/229 83% (78–87) | 188/239 79% (73–83) | 41/50 82% (69–90) | 2045/2267 90% (89–91) | – |
| 1 seropositive | 41/860 5% (4–6) | 52/889 6% (4–8) | 29/229 13% (9–18) | 38/230 16% (12–21) | 5/50 10% (4–21) | 165/2267 7% (6–8) | – |
| Over 1 seropositive | – | 30/889 3% (2–5) | 10/229 4% (2–8) | 13/239 5% (3–9) | 4/50 8% (3–19) | 57/2267 3% (2–3) | – |
| **INDIVIDUALS** | | | | | | | |
| **Age** | | | | | | | |
| 5–9 | – | 0/6 0% (0–39) | 1/38 3% (0–13) | 5/97 5% (2–12) | 2/26 8% (2–24) | 8/167 5% (2–9) | 0.5 (0.2–1.0) |
| 10–19 | – | 2/21 10% (3–29) | 8/99 8% (4–15) | 14/248 6% (3–9) | 7/91 8% (4–15) | 31/459 7% (5–9) | 0.7 (0.5–1.1) |
| 20–49 | 14/227 6% (4–10) | 39/361 11% (8–14) | 23/249 9% (6–13) | 36/375 10% (7–13) | 7/90 8% (4–15) | 119 /1302 9% (8–11) | Ref |
| 50–64 | 17/316 5% (3–8) | 38/607 6% (5–8) | 18/253 7% (5–11) | 22/224 10% (7–14) | 1/43 2% (0–12) | 96/1443 7% (5–8) | 0.7 (0.5–0.9) |
| 65+ | 10/317 3% (2–6) | 33/783 4% (3–6) | 1/48 2% (0–11) | 0/12 0% (0–24) | 0/3 0% (0–56) | 44/1163 4% (3–5) | 0.4 (0.3–0.6) |
| **Sex** | | | | | | | |
| Female | 28/558 5% (3–7) | 40/900 4% (3–6) | 27/364 7% (5–11) | 34/475 7% (5–10) | 8/135 6% (3–111) | 137/2432 6% (5–7) | Ref |
| Male | 13/302 4% (3–7) | 72/878 8% (7–10) | 24/323 7% (5–11) | 43/481 9% (7–12) | 9/118 8% (4–14) | 161/2102 8% (7–9) | 1.4 (1.1–1.8) |
| **Self-reported symptom** | | | | | | | |
| Asymptomatic or seronegative | 7/602 1% (1–2) | 36/1277 3% (2–4) | 19/449 4% (3–7) | 24/643 4% (3–5) | 5/176 3% (1–6) | 91/3147 3% (2–4) | Ref |
| Symptomatic | 34/258 13% (10–18) | 76/501 15% (12–19) | 32/238 13% (10–18) | 53/313 17% (13–21) | 12/77 15% (9–25) | 207/1387 15% (13–17) | 5.9 (4.6–7.6) |
| **Reduced contact[a]** | | | | | | | |
| No | 8/71 11% (6–21) | 1/70 1% (0–8) | 5/37 14% (6–28) | 3/39 8% (3–20) | 0/7 0% (0–35) | 17/224 8% (5–12) | – |
| Yes | 33/788 4% (3–6) | 107/1672 6% (5–8) | 40/569 7% (5–9) | 63/707 9% (7–11) | 11/178 6% (3–11) | 254/3914 6% (6–7) | 0.8 (0.5–1.5) |
| Missing Response | 0/1 0% (0–95) | 4/36 11% (4–25) | 6/81 7% (3–15) | 11/210 5% (3–9) | 6/68 9% (4–18) | 27/396 7% (5–10) | – |
| **Number of extra-HH contacts/week[b]** | | | | | | | |
| 0 | 3/64 5% (2–13) | 14/188 7% (4–12) | 7/72 10% (5–19) | 5/88 6% (2–13) | 1/12 8% (0–35) | 30/424 7% (5–10) | 0.9 (0.6–1.4) |
| 1–2 | 10/207 5% (3–9) | 26/375 7% (5–10) | 7/134 5% (3–10) | 15/180 8% (5–13) | 2/49 4% (1–14) | 60/945 6% (5–8) | 0.8 (0.5–1.2) |
| 3–5 | 12/283 4% (2–7) | 32/563 6% (4–8) | 7/158 4% (2–9) | 12/152 8% (5–13) | 3/47 6% (2–17) | 66/1203 5% (4–7) | Ref |
| 6–10 | 10/115 9% (5–15) | 22/266 8% (6–12) | 8/86 9% (5–17) | 12/132 9% (5–15) | 1/26 4% (0–10) | 53/625 8% (7–11) | 1.2 (0.8–2.0) |
| Over 10 | 6/190 3% (1–7) | 14/350 4% (2–7) | 16/156 10% (6–16) | 22/194 11% (8–17) | 4/51 8% (3–18) | 62/941 7% (5–8) | 0.9 (0.6–1.5) |
| Missing Response | 0/1 0% (0–95) | 4/36 11% (4–25) | 6/81 7% (3–15) | 11/210 5% (3–9) | 6/68 9% (4–18) | 27/396 7% (5–8) | – |
| Overall | 41/860 5% (4–6) | 112/1778 6% (5–8) | 51/687 7% (6–10) | 77/956 8% (6–10) | 17/253 7% (4–10) | 298/4534 7% (6–7) | – |

[a]A self-assessment of whether the participants have reduced the number of people they meet since the start of the epidemic.
[b]Average number of people participants meet outside of the people they lived with since the start of the epidemic.

the age of the infector were not well supported by the data (ΔWAIC −15.5 to −24.7) and included no significant differences between ages (Table S2).

Males were more likely to be infected outside (OR=1.4, 95% CrI 1.0–2.0), and possibly inside the household (OR=1.4, 95%CrI 0.6–3.1), though the latter estimate is less strongly supported by the data (Fig. 3 and Table S2).

Seropositive household members not reporting symptoms had 0.31 times the odds (95%CrI: 0.11–0.68) of infecting another household member compared to those reporting symptoms consistent with COVID-19 (Fig. 3). This difference was larger (OR=0.24, 95%CrI 0.09–0.54) when only considering those who reported symptoms more than two weeks before blood draw as symptomatic infections (Table S6, Fig. S6).

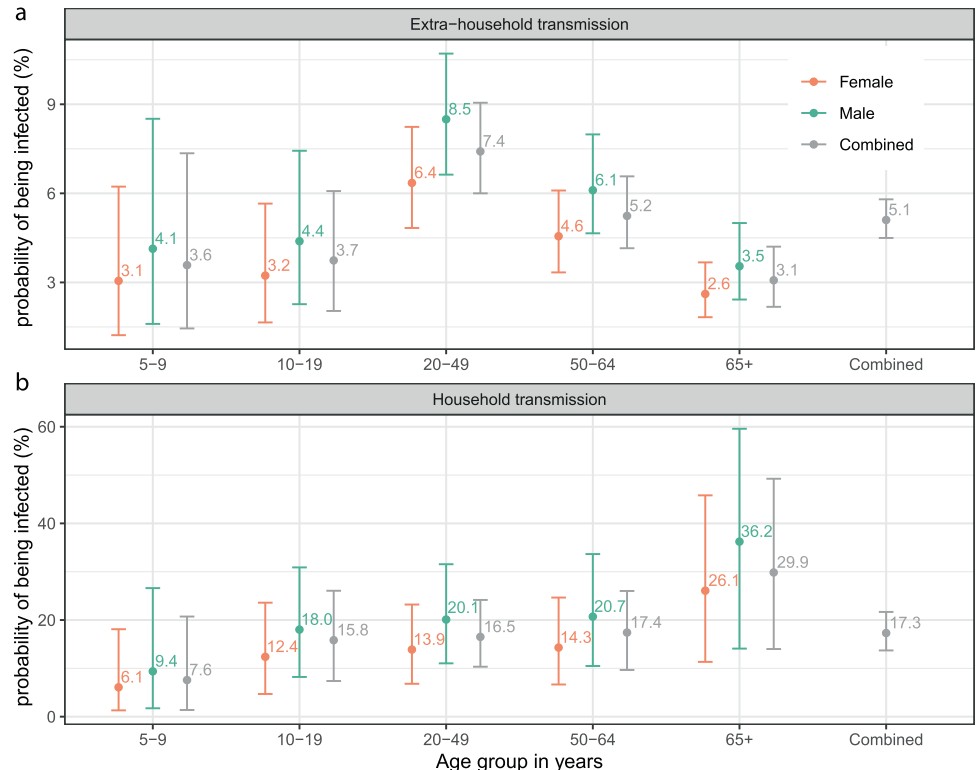

**Fig. 2 Risk of extra-household transmission and within-household transmission from a single infected household member. a** Estimated median probability of extra-household infection from the start of the epidemic in Geneva until the time of the serosurvey by age group and sex. **b** Estimated median probability of infection from a single infected household member by age group and sex. Dots and bars represent median and 95% credible intervals of the posterior distribution. Probabilities of being infected by sex and age group of the exposed individuals are estimated by a model only including age and sex of the exposed individuals (model 2, orange/green bars; see Table S2). Probabilities of being infected by the age group of the exposed individuals combining males and females (left four gray bars on both panels) are estimated with an age-only model (model 1). The overall probabilities of being infected (rightmost gray bar on both panels) are estimated with the null model (model 0).

Using posterior distributions of parameters, we simulated the source of infection for all individuals in the study. We estimate that 22.5% (95%CrI 20.1–24.2) of all infections were caused by another household member, with the proportion of infections attributable to household transmission increasing with household size (Table S3, Figs. S8 and S9). A larger proportion of infections were attributable to household transmission for those recruited after mid-May (last 6 weeks of the study, 27.8, 95%CrI 22.8–30.4) compared to those recruited in the first six weeks of the study (first 6 weeks: 20.5, 95%CrI 17.8, 22.4). In households with two individuals, 23.2% (95%CrI 19.6–25.9) of infections were between household members, increasing to 41.2% (95%CrI 29.4–47.1) in households of five people (Table S3). Of within-household infections, we estimate 14.5% (95%CrI 7.2–22.7) were due to individuals not reporting symptoms consistent with COVID-19.

Here we focus on the results of the best fitting models, but across the ten models considered (Table S2), estimates were qualitatively and quantitatively consistent with the primary findings. Similarly, we explored the sensitivity of our results to the ELISA seropositivity cutoff and found no qualitative differences in results (Fig. S4).

### Discussion

The results presented here appropriately place symptomatic household transmission of SARS-CoV-2 in the context of community risk and asymptomatic spread. We show an approximate 1 in 6 risk (17.3%) of being infected by a single SARS-CoV-2 infected household member (Table S3). This contrasts with a 1 in 20 chance (5.1%) of being infected in the community over most of

the first epidemic wave in Geneva, a period of roughly 2 months. Despite the high risk of transmission from an infected household member, as in many cities in high-income nations, households are mostly small limiting opportunities for onward transmission. Thus, less than a quarter of cases could be attributed to transmission between household members. While asymptomatic individuals appear to be less than a third as likely to transmit, they cannot be dismissed as inconsequential to disease spread, and are responsible for one in six within-household transmissions in this study. Our results are suggestive of the dual roles of biology and social behavior in shaping age-specific infection patterns, with the age signature of risk within households indicative of lower biological susceptibility in the very young, and elevated susceptibility in the old; while extra-household risk seems more driven by behavior, with working age adults being at the highest risk.

It has long been thought that asymptomatic individuals are less likely to transmit than symptomatic ones, though studies have recovered similar concentrations of viral RNA from nasopharyngeal samples from these two groups[15]. By using serological data, we were able to show that those not reporting symptoms have one-third the odds of transmitting within households as symptomatic ones, similar to a study from Wuhan, China[16], and ultimately caused about 15% of household infections. This reduced transmissibility may be due to reduced duration of viral shedding and reduced ability to mechanically spread virions (e.g., through coughs). We did not assess the role of asymptomatics in community spread, but it is plausible that they may play an even larger role there, as symptomatic

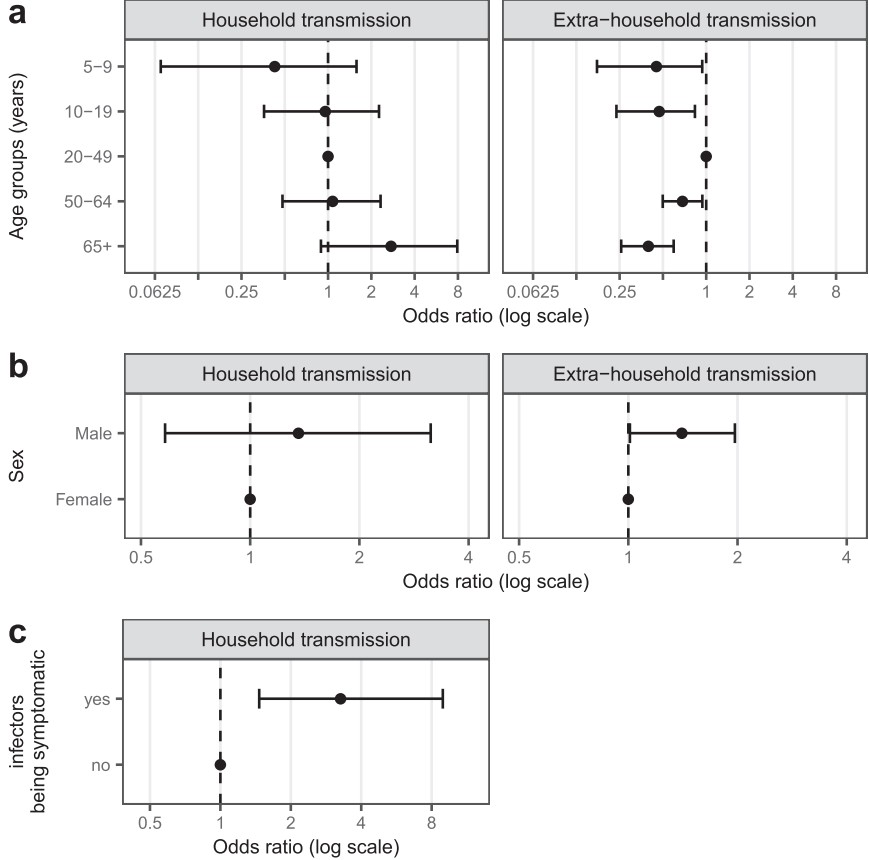

**Fig. 3 Risk factors of SARS-CoV-2 infection and transmission.** Relative odds of being infected outside the household and from a single infected household member by individual characteristics of the exposed individuals, **a** age group, **b** sex, and **c** potential infectors' symptom status. Odds ratios and credible intervals, shown on the log-scale, are estimates from model 4 (see Table S2).

individuals are more likely to stay home or take extra precautions to reduce exposures when sick.

As with previous studies of SARS-CoV-2 transmission among household members and other close contacts[2,17,18], we find evidence supporting a reduced risk of infection from household exposures among young children, and elevated risk of infection among those 65 or older. However, it is important to note that we only find this reduced risk among the youngest children in our study (5–9-year-olds), while 10–19 year olds have a similar risk profile to working age adults. The other PCR-based household study that reported per-exposure transmission did not report susceptibility results from this age group[10]. This is consistent with the hypothesis that young children may be biologically less susceptible to SARS-CoV-2 infection, though heterogeneity in social contact and other behaviors within households cannot be ruled out.

Patterns of extra-household infection suggest social factors dominate this risk, as both young children and older adults are at reduced risk of infection compared to working age adults. As children have returned to schools in Geneva (mid-May 2020), the social factors driving this pattern have likely changed significantly and we may see children become a more significant source of extra-household infections[19], despite their apparently lower susceptibility. The risk that infected young children pose to their household members is unclear; the sample size was likely too low to detect small to moderate differences in risk. While there are mixed results in the literature on age-specific differences in infectiousness[20], a large study from Wuhan, China suggested that those less than 20 years old are more likely to infect others than adults 60 years and older, given the same amount of exposure[16].

We did not find any significant relationship between the age of an infector and probability of transmission (nor did including these terms improve model fit), but children are less often symptomatic[21] and we did find a strong relationship between symptoms and transmission.

Our study has a number of important limitations. Symptoms were self-reported and, given that the times of infection are unknown, they may not necessarily have been a result of the SARS-CoV-2 infection. We cannot exclude recall bias in symptom reports and other self-reported exposures. Further, we looked at only a narrow range of symptoms to increase specificity, which left out more general potentially SARS-CoV-2-related symptoms (e.g., nausea, diarrhea). We detected only eight seropositive children under the age of 10, leading to large uncertainty in age-specific risk estimates for this group. Although extra-household estimates are informed by data from all households, within-household estimates are only informed by data from households with at least one seropositive member (222/2267 households), thus limiting our statistical power. While validation data of the Euroimmun ELISA from across the world have confirmed its high specificity and sensitivity for detecting recent infections[22–24], most data are from adults, and it is possible that performance in young children may be different. Most of the participants in the study were recruited after the epidemic peak and it is possible that we did not fully capture all infections in each household due to insufficient time to mount a detectable response. Conversely, we may have also missed infections due to waning of responses. However, antibody responses appear to generally sustain over the first 4 months from infection, the plausible infection time window of participants in this study[25]. When conducting stratified

analyses including households recruited early and late, we found few qualitative differences in the primary results (Figs. S5 and S7). We included only households where all household members provided blood samples in the main analysis, but sensitivity analyses of all enrolled individuals led to similar primary results (Table S6). Given the cross-sectional nature of our data, all transmission chains within households were equally likely within our modeling framework, which led to larger uncertainty than having prospectively collected data. However, collection of these data over thousands of households can be challenging, and we show that more commonly collected data from serosurveys can be leveraged to refine our understanding of transmission.

This study captures infections that occurred during the first wave of the pandemic in Geneva, a period of time when workplaces and schools were largely closed and peoples' social contacts were greatly reduced. In future phases of this pandemic, when social contact patterns change the proportion of transmission that occurs between household members and potentially age-sex specific risks could differ. While we found no evidence in previous analyses of these data for differences in seropositivity by neighborhood wealth or education[26], these and other indicators of wealth might be associated with transmission risk within Geneva or in other populations. Likewise, the general nature of the Geneva population and the control measures in place may limit the generalizability of our estimates of absolute risk of infection, attributable fraction, and extra-household risks. For example, the increasing importance of household transmission with increasing household size (Fig. S8) suggests household transmission would be far more important in settings with larger households. However, we believe our estimates of relative risks by age and symptom status within households, which are likely more biologically driven, should be generalizable to most settings; as should our general observations about how social and biological factors influence different types of transmission.

Our study highlights how biological and social factors might combine to shape the risk of SARS-CoV-2 infection. While we expect some differences across settings, we believe that the general trend in per-exposure infection risk by age and sex and increased infectiousness of symptomatic individuals are fundamental attributes of this pandemic. These differences have important implications for guiding patient care and public health policy. For example, increased susceptibility of the oldest individuals suggests that rapid and aggressive measures are needed to protect them as soon as there is any possibility that SARS-CoV-2 was introduced into their living environment. At the population level, quantifying the infectiousness of asymptomatics can help us understand the extent the pandemic is driven by asymptomatic infections. Our study provides a model for using cross-sectional serologic surveys to assess the relative contribution of household and community transmission. As countries continue to alter quarantine and self-isolation policies, disentangling the contribution of household and community transmission can help evaluate success of these intervention strategies. Continued serological and virologic monitoring of diverse populations with detailed analyses like those presented here are critical to the continued evidence-based response to this pandemic.

## Methods

*Study design, participants, and procedures.* The SEROCoV-POP study is a cross-sectional population-based survey of former participants of an annual survey of individuals 20–74 years old representative of the population of Geneva (Canton), Switzerland. The enrollment into the study occurred from April through June 2020 during the first wave of the SARS-CoV-2 pandemic in Geneva. First wave lockdown measures (including school closures) started in mid-March and largely ended by the end of May. The full survey protocol is available online and a detailed description of the design and seroprevalence results were previously published[11,26].

The SEROCoV-POP study invited all 10,587 participants of the previous annual surveys to participate in the study through email or post. Participants were invited to bring all members of their household aged 5 years and older to join the study. After providing informed written consent, participants either filled out a questionnaire online, in the days before their visit, or at the time of their visit at one of two enrollment locations (the main canton hospital and one satellite location) within Geneva. The questionnaire included questions about participants' demographics, household composition, symptoms since January 2020, details on the frequency of extra-household contacts and reduction in social interaction since the start of the pandemic. Only participants 14 years and older were asked about their frequency of extra-household contacts and changes in behavior. Despite this age cut off, we use more standard age cutoffs (10–19 years) in our analysis for comparability with other studies[11]. We defined symptom presentation a priori as having reported any of: cough, fever, shortness of breath, or loss of smell or taste since January 2020 (symptoms reported in the 2-week prior to testing were excluded in a sensitivity analysis). We collected peripheral venous blood from each consenting participant. Households where all members provided blood samples were included in the present analysis (there was a 100% questionnaire response rate in this group). As blood was not collected from children under 5, all households with children in this age group were excluded. We conducted a sensitivity analysis with all households, regardless of whether all members provided blood samples, effectively treating household members outside the study as a community source of infection. All participants gave written informed consent before participation in the SEROCoV-POP study. For individuals younger than 18 years, parents or a legal representative provided consent. The study was approved by the Cantonal Research Ethics Commission of Geneva, Switzerland (CER16-363).

*Laboratory analysis.* We assessed anti-SARS-CoV-2 IgG antibodies in each participant using an ELISA (Euroimmun; Lübeck, Germany #EI 2606-9601 G) targeting the S1 domain of the spike protein of SARS-CoV-2; sera diluted 1:101 were processed on a EuroLabWorkstation ELISA (Euroimmun). An in-house validation study found that the manufacturer's recommended cutoff for positivity ($\geq 1.1$) had a specificity of 99% and sensitivity of 93%, based on positive controls tested between 0 and 39 days after symptom onset[24]. In our primary analyses we defined seropositivity based on the cutoff recommended by the manufacturer and explored a higher cut-off of 1.5 (>1.5) in sensitivity analyses[24]. As the presence of antibodies has been shown to be a reliable marker of past infection, we use the term "infected" to refer to a seropositive individual.

*Statistical analyses.* We fit chain binomial transmission models to estimate two primary quantities; the average probability of extra-household infection from the start of the epidemic through the time of blood draw across Geneva (referred to also as "community infections" over the first epidemic wave) and the probability of being infected from a single infected household member over the course of his/her infectious period (referred to as "household exposures"; see supplemental text for model assumptions)[27,28]. We assume that serologic status is a perfect marker of having been infected, that individuals cannot get reinfected, and that all individuals were susceptible at the start of the pandemic. When fitting these models we explicitly consider all possible sequences of viral introductions to each household and subsequent transmission events within the household. For example, in a household

with 2 seropositive individuals, both could have been infected outside of the household, or one could have been infected outside and then infected one other person within the household. We adapted models to estimate the within household and extra-household transmission risk according to the characteristics of potential infectees (age, sex, self-reported extra-household contact behavior) and, for within-household risk, those of the potential infectors (symptoms, age). As extra-household contact questions were only asked to those over 14 years old, we compared extra-household transmission by self-reported reduction or frequency in social contacts only for those 20 years and older. We imputed a small number of missing data (1%, 36/3908) related to extra-household contacts among those who were 20 years and older based on household averages (see supplement). We simulate the proportion of infections attributable to extra-household and within household exposures.

We built a series of ten models including different combinations of individual-level characteristics (e.g., age, sex, self-reported contacts, symptoms) and compared their fit using the widely applicable information criterion (WAIC)[29]. We implemented the models in the Stan probabilistic programming language and used the *rstan* package (version 2.21.0) to sample from the posterior distribution and analyse outputs[30]. We used weakly informative priors on all parameters to be normally distributed on the logit scale with mean of 0 and standard error of 1.5. We ran four chains of 1,000 iterations each with 250 warm-up iterations and assessed convergence visually and using the Gelman-Rubin Convergence Statistic (R-hat)[31]. All estimates are means of the posterior samples with the 2.5th and 97.5th percentiles of this distribution reported as the 95% credible interval. Full model and inference details are provided in the supplement and code needed to reproduce analyses are available at https://github.com/HopkinsIDD/serocovpop-households (https://doi.org/10.5281/zenodo.4740044).

**Reporting summary**. Further information on research design is available in the Nature Research Reporting Summary linked to this article.

## Data availability
Data can be made available to share upon submission of a data request application to the investigators board via the corresponding author or S.S. (silvia.stringhini@hcuge.ch). Data needed for testing the code can be found at https://github.com/HopkinsIDD/serocovpop-households (https://doi.org/10.5281/zenodo.4740044).

## Code availability
All relevant code can be found at https://github.com/HopkinsIDD/serocovpop-households.

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

## Acknowledgements
Funding was provided by Swiss Federal Office of Public Health, Swiss School of Public Health (Corona Immunitas research program), Fondation de Bienfaisance du Groupe Pictet, Fondation Ancrage, Fondation Privée des Hôpitaux Universitaires de Genève, and Center for Emerging Viral Diseases.

## Author contributions

A.S.A. and S.S. had full access to all of the data in the study and took responsibility for the integrity of the data and the accuracy of the data analysis. A.S.A. and J.L. have contributed equally. Concept and design: Q.B., J.L., S.S., A.S.A. Acquisition, analysis, or interpretation of data: Q.B., J.L., I.E., S.A.L., L.K., N.V., D.A.T.C., A.F., D.P., I.G., S.S., A.S.A. Drafting of the manuscript: Q.B., J.L., S.S., A.S.A. Critical revision of the manuscript for important intellectual content: Q.B., J.L., I.E., S.A.L., L.K., N.V., D.A.T.C., A.F., D.P., I.G., S.S., A.S.A.; Statistical analysis: Q.B., J.L., S.A.L., A.S.A. Obtained funding: S.S., I.G. Administrative, technical, or material support: S.S., I.G., D.P., L.K., N.V. Supervision: ASA, S.A.L., SS, L.K.

## Competing interests

The authors declare no competing interests.

## Additional information

## SEROCoV-POP Study Group

Silvia Stringhini[10,12,13], Idris Guessous[10,12], Andrew S. Azman [1,11,12,24✉], Hélène Baysson[15], Prune Collombet[12,15], David De Ridder[15], Paola d'Ippolito[12], Matilde D'asaro-Aglieri Rinella[12], Yaron Dibner[12], Nacira El Merjani[12], Natalie Francioli[12], Marion Frangville[15], Kailing Marcus[12], Chantal Martinez[12], Natacha Noel[12], Francesco Pennacchio[12], Javier Perez-Saez[1,4], Dusan Petrovic[12,13], Attilio Picazio[12], Alborz Pishkenari[12], Giovanni Piumatti[12,16], Jane Portier[12], Caroline Pugin[12], Barinjaka Rakotomiaramanana[12], Aude Richard[11,12], Lilas Salzmann-Bellard[12], Stephanie Schrempft[12], Maria-Eugenia Zaballa[12], Zoé Waldmann[15], Ania Wisniak[11], Alioucha Davidovic[15], Joséphine Duc[15], Julie Guérin[15], Fanny Lombard[15], Manon Will[15], Antoine Flahault[11,12,15], Isabelle Arm Vernez[2], Olivia Keiser[11], Loan Mattera[17], Magdalena Schellongova[15], Laurent Kaiser[2,4,6,15], Isabella Eckerle[2,6,15], Pierre Lescuyer[6], Benjamin Meyer[15,18], Géraldine Poulain[6], Nicolas Vuilleumier[6,15], Sabine Yerly[2,6], François Chappuis[12,15], Sylvie Welker[12], Delphine Courvoisier[12], Laurent Gétaz[12,15], Mayssam Nehme[12], Febronio Pardo[19], Guillemette Violot[20], Samia Hurst[21], Philippe Matute[12], Jean-Michel Maugey[19], Didier Pittet[22], Arnaud G. L'Huillier[15,23], Klara M. Posfay-Barbe[15,23], Jean-François Pradeau[19], Michel Tacchino[19] & Didier Trono[11]

[15]Faculty of Medicine, University of Geneva, Geneva, Switzerland. [16]Faculty of BioMedicine, Università della Svizzera italiana, Lugano, Switzerland. [17]Campus Biotech, Geneva, Switzerland. [18]Centre for Vaccinology, Department of Pathology and Immunology, University of Geneva, Geneva, Switzerland. [19]Information Systems Division, Geneva University Hospitals, Geneva, Switzerland. [20]Division of Communication, Geneva University Hospitals, Geneva, Switzerland. [21]Institut Ethique, Histoire, Humanités, University of Geneva, Geneva, Switzerland. [22]Infection Prevention and Control Program and World Health Organization (WHO) Collaborating Centre on Patient Safety, Geneva University Hospitals, Geneva, Switzerland. [23]Division of General Pediatrics, Geneva University Hospitals, Geneva, Switzerland.

