## [Peer Review File · Nature Communications]

REVIEWER COMMENTS

Reviewer #1 (Remarks to the Author):

This paper reports the risk of SARS-CoV-2 infection associated with intra- vs. extra-household exposures using data from a cross-sectional survey in Geneva, Switzerland early in the COVID-19 pandemic. A large number of studies have already reported a wide range of household transmission estimates, but most detected infection with PCR, which can miss infection if not conducted early after exposure. This study performed serology in a large cross-sectional sample. Serology may have higher sensitivity to detect total infections over a study period. The study also included a large sample size than other prior studies assessing household transmission. However, because a cross-sectional design was used, information about the relative timing of infection among household members was not observed. The availability of public, well-documented analysis scripts strengthens the reproducibility of this work.

The paper would benefit from a more thorough discussion of the model's assumptions, which were only listed in the supplement. I suggest that the authors include more details about the assumptions of the model in the main text and better justify them. For example, the models assumed that the initial population was completely susceptible. The study period started in April 2020, so it is possible that a small proportion of participants were not susceptible at baseline. The previous paper reporting the SEROCov-POP study in the Lancet shows that cumulative cases were in the thousands by early April. Please include additional details about the estimated incidence (or at least confirmed case counts) at the beginning of the study period to justify this assumption and discuss how the presence of individuals who were no longer susceptible in the population may have affected findings (i.e., in which direction).

An additional assumption of the chain binomial model that was not mentioned is that extra-household sources of infection are distributed evenly in the community. Please justify this assumption as well and discuss how it might affect model outputs.

Please also discuss the limitations of cross-sectional data in more detail in the limitations section. Prospective data including the time of infections, as is used in many other household transmission studies, would have allowed the authors to use a modeling approach that imposed fewer assumptions.

Minor comments

1. In the limitations of the abstract, I suggest mentioning the lack of information on the timing of infection due to the cross-sectional design.
2. More description of the non-pharmaceutical interventions and policies in place at the time of the study would help readers contextualize results and assess external validity. For example, were schools and businesses closed at the time?
3. Related to the last point, on line 289: "As children have returned to school in Geneva..." When? Since the study was completed?
4. Even though the survey protocol was previously reported, please add at least a few more sentences to the methods to describe the survey. For example, on line 129 it is not clear what site is referred to. Please also state the period during which participants were invited to the survey for this study so that readers can understand the length of recall once they reach line 131.
5. Not all supplemental figures/tables are referenced in text (e.g., Supplement Table 5).
6. Figure 2 would be easier to read if the width of the position dodge (space between x axis categories) was smaller.
7. In the tables (or table captions), please define reduced contact. Was this at any time during the recall period?

8. Was information on employment type and education level available? If so, please include it in table 1. This information would help readers assess the generalizability of findings to other populations.

9. Line 260 makes a statement about the proportion of cases that can be attributed to household transmission. Please include more details about the representativeness of the survey to the target population in the results section to support this inference. This section only compares the distribution of age and household size in the study sample to the general population.

10. Line 264: "Our results further illustrate the dual roles of biology and social behavior in shaping age-specific infection patterns..." A similar statement is made on line 330. While I agree with the gist of these sentences, this study can't distinguish between these factors since very little data was collected on behaviors. Please revise this to use language to indicate that results were suggestive of these roles rather than illustrative of them.

11. Line 331: With regard to the statement on general trends in household transmission by age and sex, this could vary substantially if assessed during a period when schools, universities, and workplaces were open. Please at least mention this possibility here.

12. Please add more discussion of how the results compare to other household transmission studies, discussing how the results may differ for those that used PCR instead of serology (summarized in Madewell et al) or those that used serology and had information about the timing of infection (e.g., Lewis et al. DOI:10.1093/cid/ciaa1166.)

13. I suggest that the last paragraph touch on the relative contribution of household vs. external transmission – a key focus of the paper – and the policy relevance of this finding.

Reviewer #2 (Remarks to the Author):

I read with interest the paper by Bi et al. on the household transmission of SARS-CoV-2. This is a modelling study using population-based serological survey in Geneva. The authors included 2000+ households between April and June 2020 and confirmed history of SARS-CoV-2 infection through IgG. The authors highlighted that the probability of intra-household infection was much higher than that of extra-household infection (17.3% vs 5.1%), and that young children had a lower risk of infection from household members. Based on self-reported history of COVID-19 symptoms, the authors concluded that asymptomatic infections are far less likely to transmit than symptomatic infections but do cause infections.

Overall it is an interesting study with a novel modelling framework that is based on assumptions regarding generations of intra-household transmission. However, I have several major concerns, most regarding the statistical power of the study in the light of this modelling framework, as detailed below.

Major comments:

- It was not explicitly stated in the main text what proportion of the 2267 households were actually used by the model. My understanding is that the modelling framework can only incorporate data from households with at least one IgG positive member — 9.8% (222/2267). The majority of the findings in this study actually came from this subset of 222 households. The authors should make this clear if this is the case, and acknowledge this as a very key limitation in the abstract (under "limitations") and discussion.
- The small amount of data included (i.e. 222 households) largely explained why a great amount of the parameter estimates had wide credible intervals. In fact, some of the stated findings seemed to be based on the point estimate. For example, in lines 66-68 of the abstract, the odds of infection for the age groups all include 1, meaning that we were not yet there to say the odds of infection in $\geq 65y$ were different from the reference group (i.e. 20-49y). The authors should check this point throughout the manuscript. For example, they might state that there was an

increasing trend over age in the odds of infection although being not statistically significant.

- There has been growing interest in the role of children in bringing the infection to others, i.e. infectivity. The authors incorporated infectivity in the model although showing that this did not improve the model and did not show age-specific differences. Is it due to the lack of the statistical power? Furthermore, the authors are encouraged to comment on the findings from a recent household transmission study in Wuhan, which found that children had higher infectivity. ([https://www.thelancet.com/journals/laninf/article/PIIS1473-3099\(20\)30981-6/fulltext](https://www.thelancet.com/journals/laninf/article/PIIS1473-3099(20)30981-6/fulltext))

Minor comments:

- Line 75: "aged ≥ 5 years" would be clearer.
- Lines 104–106: the authors seemed to suggest that serologic studies are better than virologic studies in answering the questions regarding household transmission. I think it would be more appropriate to state that they complement each other rather than saying one is superior to the other, since virologic data has its own weaknesses.
- Line 609: SARS-CoV-2.

Reviewer #3 (Remarks to the Author):

Summary

In their work, the authors use insights from a population-based serological survey and questionnaire to estimate intra and extra household transmission rates. Whereas I believe there is quite some potential in this work, I do have several concerns with the analysis as it is now.

The authors present a complete case analysis as the main analysis. I would argue that the main analysis should take into account all available data. I would also encourage the authors to discuss the assumed missingness process behind their analyses. Certainly given that descriptive statistics about the two samples highlights important differences between both. The authors write on line 170 that imputation was based on household averages; please clarify.

The manuscript would benefit from having a clear view on the intervention measures taken during the study period. The question is also whether these intervention measures had any impact on the findings presented in this study. More in general, I would love to see some temporal sensitivity analyses given changes possibly due to intervention measures, waning of antibodies as observed in other studies where the Euroimmun assay was used and differences in terms of behavior because of an overlap of the epidemic curve and the study period in which it is clear that household composition changed (see S Table 1).

Although effect sizes are important, at times, I do think the authors are too confident in reporting effect sizes whereas these effects are not significant. More careful wording is needed in these instances (eg lines 214-217).

Can the authors clarify why they selected model 5 and not model 6? If the authors, as indicated in the methodology section go for selection based on WAIC, why don't they follow this here? This, to me, looks like a very arbitrary way of selecting a model.

REVIEWER COMMENTS

Reviewer #1 (Remarks to the Author):

This paper reports the risk of SARS-CoV-2 infection associated with intra- vs. extra-household exposures using data from a cross-sectional survey in Geneva, Switzerland early in the COVID-19 pandemic. A large number of studies have already reported a wide range of household transmission estimates, but most detected infection with PCR, which can miss infection if not conducted early after exposure. This study performed serology in a large cross-sectional sample. Serology may have higher sensitivity to detect total infections over a study period. The study also included a large sample size than other prior studies assessing household transmission. However, because a cross-sectional design was used, information about the relative timing of infection among household members was not observed. The availability of public, well-documented analysis scripts strengthens the reproducibility of this work.

The paper would benefit from a more thorough discussion of the model's assumptions, which were only listed in the supplement. I suggest that the authors include more details about the assumptions of the model in the main text and better justify them. For example, the models assumed that the initial population was completely susceptible. The study period started in April 2020, so it is possible that a small proportion of participants were not susceptible at baseline. The previous paper reporting the SEROCov-POP study in the Lancet shows that cumulative cases were in the thousands by early April. Please include additional details about the estimated incidence (or at least confirmed case counts) at the beginning of the study period to justify this assumption and discuss how the presence of individuals who were no longer susceptible in the population may have affected findings (i.e., in which direction).

The primary model assumptions related to susceptibility are that serologic status is a perfect marker for susceptibility and that all individuals were susceptible at the start of the pandemic. We do not need to assume that all are susceptible in April 2020, when this cross-sectional study began since serology captures events in the past (unlike prospective household studies based on virologic confirmation, for example). In our model we consider all potential infection chains for positive individuals, including that each was infected by household contacts or by extra-household contacts with no further assumptions about susceptibility. We have added the following:

"We assume that serologic status is a perfect marker of having been infected; that individuals cannot get infected twice, and that all individuals were susceptible at the start of the pandemic."

An additional assumption of the chain binomial model that was not mentioned is that extra-household sources of infection are distributed evenly in the community. Please justify this assumption as well and discuss how it might affect model outputs.

We are estimating the average probability of extra-household infections across all individuals in this study. While risk certainly varies across individuals, time and from location-to-location,

based on weekly sitreps from the Canton of Geneva, it is clear that SARS-CoV-2 incidence was geographically widespread with no clear areas of persistent elevated incidence across the epidemic (e.g., <https://www.ge.ch/document/19696/annexe/35>). We also tried to account for some of the potential differences in extra-household risk by including self-reported extra-household contacts into the model of this probability. We clarified that the model captures the average probability of extra-household infection in the methods:

“We fit chain binomial transmission models to estimate two primary quantities; the average probability of extra-household infection from the start of the epidemic through the time of blood draw across Geneva (referred to also as ‘community infections’ over the first epidemic wave) and the probability of being infected from a single infected household member over the course of his/her infectious period (referred to as ‘household exposures’; See supplemental text for model assumptions).”

Please also discuss the limitations of cross-sectional data in more detail in the limitations section. Prospective data including the time of infections, as is used in many other household transmission studies, would have allowed the authors to use a modeling approach that imposed fewer assumptions.

We agree with the reviewer that prospective data would strengthen our analyses and results. We added the following discussion to the limitation section:

“Given the cross-sectional nature of our data, all transmission chains within households were equally likely within our modeling framework, which led to larger uncertainty than having prospectively collected data. However, collection of these data over thousands of households can be challenging, and we show that more commonly collected data from serosurveys can be leveraged to refine our understanding of transmission.”

Minor comments

1. In the limitations of the abstract, I suggest mentioning the lack of information on the timing of infection due to the cross-sectional design.

We rewrote the abstract to comply with the Nature Communications’ format, which typically does not have listed limitations. However, we did edit the limitation section to mention the lack of information on the timing of infection. Please see the quote in reply to the third comment of reviewer 1.

2. More description of the non-pharmaceutical interventions and policies in place at the time of the study would help readers contextualize results and assess external validity. For example, were schools and businesses closed at the time?

We have added a description of the timelines of lockdown measures including school closures in the method:

“The enrollment into the study occurred from April through June 2020 during the first wave of the SARS-CoV-2 pandemic in Geneva. First wave lockdown measures (including school closures) started in mid March and largely ended by the end of May.”

3. Related to the last point, on line 289: “As children have returned to school in Geneva...” When? Since the study was completed?

We have added the timing of school openings to this line (mid-May 2020).

4. Even though the survey protocol was previously reported, please add at least a few more sentences to the methods to describe the survey. For example, on line 129 it is not clear what site is referred to. Please also state the period during which participants were invited to the survey for this study so that readers can understand the length of recall once they reach line 131.

We have updated the Methods section to include a few more details about the study as suggested:

“The enrollment into the study occurred from April through June 2020 during the first wave of the SARS-CoV-2 pandemic in Geneva. First wave lockdown measures (including school closures) started in mid March and largely ended by the end of May.”

“After providing informed written consent, participants either filled out a questionnaire online, in the days before their visit, or at the time of their visit at one of two enrollment locations (the main canton hospital and one satellite location) within Geneva.”

5. Not all supplemental figures/tables are referenced in text (e.g., Supplement Table 5).

We have gone through the main text and captions and made sure all supplemental figures and tables are now references. We referenced Table S5 in the caption of Figure S1. *“Comparison of the characteristics of those excluded due to being in an incomplete household to those included in the main analysis are shown in Table S5.”*

6. Figure 2 would be easier to read if the width of the position dodge (space between x axis categories) was smaller.

We edited the figure based on the reviewer's suggestion.

7. In the tables (or table captions), please define reduced contact. Was this at any time during the recall period?

We clarified the definition of this variable in the footnote of table 1.

“A self-assessment of whether the participants have reduced the number of people they meet since the start of the epidemic.”

8. Was information on employment type and education level available? If so, please include it in table 1. This information would help readers assess the generalizability of findings to other populations.

Yes, information on education level and employment type are available. We previously reported distribution of education level in Supplemental table 5, and we now added distribution of employment status in the same table.

	Study participants in complete households, % (n); N=4,534	Study participants in incomplete household, % (n); N=3,810
Employment Status*		
Retired	27.3% (1236)	10.2% (464)

Student	11.0% (498)	11.9% (540)
Employed	40.9% (1853)	44.1% (2000)
Freelance	7.9% (358)	6.8% (307)
Unemployed	6.3% (284)	6.7% (304)
Missing Response	1.0% (46)	1.4% (64)

9. Line 260 makes a statement about the proportion of cases that can be attributed to household transmission. Please include more details about the representativeness of the survey to the target population in the results section to support this inference. This section only compares the distribution of age and household size in the study sample to the general population.

We added a comment on the representativeness of education level in our study population to the general population as suggested. However, education level doesn't appear to be associated with risk of seroprevalence in Geneva, which we also point out in the discussion (<https://www.medrxiv.org/content/10.1101/2020.12.16.20248180v1.full.pdf>). We also added the distribution of education level and employment status of our study population in supplemental table 5 .

"Our study sample, like that of the original SEROCov-POP study, was more educated than the general canton population with only 8.5% not having a high school degree or equivalent, compared with 23.5% in the general canton population ¹⁹ (Table S6)."

10. Line 264: "Our results further illustrate the dual roles of biology and social behavior in shaping age-specific infection patterns..." A similar statement is made on line 330. While I agree with the gist of these sentences, this study can't distinguish between these factors since very little data was collected on behaviors. Please revise this to use language to indicate that results were suggestive of these roles rather than illustrative of them.

We edited the sentence per reviewer's suggestion:

"Our results are suggestive of the dual roles of biology and social behavior collectively shape age-specific infection patterns"

11. Line 331: With regard to the statement on general trends in household transmission by age and sex, this could vary substantially if assessed during a period when schools, universities, and workplaces were open. Please at least mention this possibility here.

We would only expect household transmission by age and sex to change if the contact rates within a household or the nature of the contacts changed significantly. However, extra-

household risk would likely change significantly if schools and workplace were open. We have discussed these in the discussion section as follows:

“This study captures infections that occurred during the first wave of the pandemic in Geneva, a period of time when workplaces and schools were largely closed and peoples’ social contacts were greatly reduced. In future phases of this pandemic, when social contact patterns change, the proportion of transmission that occurs between household members and potentially age-sex specific risks could differ. ... However, we believe our estimates of relative risks by age and symptom status within households, which are likely more biologically driven, should be generalizable to most settings.”

12. Please add more discussion of how the results compare to other household transmission studies, discussing how the results may differ for those that used PCR instead of serology (summarized in Madewell et al) or those that used serology and had information about the timing of infection (e.g., Lewis et al. DOI:10.1093/cid/ciaa1166.)

We added more discussion about how our study results compare to those of other studies that report age-specific susceptibility and transmissibility by symptom status in the discussion section.

“As with previous studies of SARS-CoV-2 transmission among household members and other close contacts^{2,22}, we find evidence supporting a reduced risk of infection from household exposures among young children, and elevated risk of infection among those 65 or older. However, it is important to note that we only find this reduced risk among the youngest children in our study (5-9 year olds), while 10-19 year olds have a similar risk profile to working age adults. The other PCR-based household study that reported per-exposure transmission did not report susceptibility results from this age group¹⁰. ”

“By using serological data, we were able to show that those not reporting symptoms have one-third the odds of transmitting within households as symptomatic ones, similar to a study from Wuhan, China²²...”

13. I suggest that the last paragraph touch on the relative contribution of household vs. external transmission – a key focus of the paper – and the policy relevance of this finding.

We added discussion on relative contribution of household vs external transmission in the last paragraph.

“Our study provides a model for using cross-sectional serologic surveys to assess the relative contribution of household and community transmission. As countries continue to alter quarantine policy, disentangling the contribution of household and community transmission help can evaluate success of these intervention strategies.”

Reviewer #2 (Remarks to the Author):

I read with interest the paper by Bi et al. on the household transmission of SARS-CoV-2. This is a modelling study using population-based serological survey in Geneva. The authors included 2000+ households between April and June 2020 and confirmed history of SARS-CoV-2 infection through IgG. The authors highlighted that the probability of intra-household infection was much higher than that of extra-household infection (17.3% vs 5.1%), and that young children had a lower risk of infection from household members. Based on self-reported history of COVID-19 symptoms, the authors concluded that asymptomatic infections are far less likely to transmit than symptomatic infections but do cause infections.

Overall it is an interesting study with a novel modelling framework that is based on assumptions regarding generations of intra-household transmission. However, I have several major concerns, most regarding the statistical power of the study in the light of this modelling framework, as detailed below.

Major comments:

- It was not explicitly stated in the main text what proportion of the 2267 households were actually used by the model. My understanding is that the modelling framework can only incorporate data from households with at least one IgG positive member — 9.8% (222/2267). The majority of the findings in this study actually came from this subset of 222 households. The authors should make this clear if this is the case, and acknowledge this as a very key limitation in the abstract (under “limitations”) and discussion.

Within household data are only informed by those with at least one positive person but the outside household risk is informed by all households, which we now discuss in the limitation section.

“Although extra-household estimates are informed by data from all households, within-household estimates are only informed by data from households with at least one seropositive member (222/2267 households), thus limiting our statistical power.”

- The small amount of data included (i.e. 222 households) largely explained why a great amount of the parameter estimates had wide credible intervals. In fact, some of the stated findings seemed to be based on the point estimate. For example, in lines 66-68 of the abstract, the odds of infection for the age groups all include 1, meaning that we were not yet there to say the odds of infection in ≥ 65 y were different from the reference group (i.e. 20-49y). The authors should check this point throughout the manuscript. For example, they might state that there was an increasing trend over age in the odds of infection although being not statistically significant.

We have reviewed the document to try to make sure we are clear where findings are not statistically significant. We note however, that statistical significance is not the only measure of the strength of evidence. We adapted the following paragraphs:

“The risk of being infected by a household member was the lowest among 5-9 years old and highest among those 65 years and older, with teenagers and working age adults sharing similar risks (Figures 2,3)... Though credible intervals on individual estimates are wide, inclusion of age substantially improved model fit (Δ WAIC -14.8, Table S2)”

“Those aged 20-64 who reported reducing extra-household contacts during the pandemic had a 33% reduction in the odds of extra-household infection (OR=0.67, 95%CrI 0.40-1.2), though this estimate had large uncertainty with the credible interval including the null.”

- There has been growing interest in the role of children in bringing the infection to others, i.e. infectivity. The authors incorporated infectivity in the model although showing that this did not improve the model and did not show age-specific differences. Is it due to the lack of the statistical power? Furthermore, the authors are encouraged to comment on the findings from a recent household transmission study in Wuhan, which found that children had higher infectivity. ([https://www.thelancet.com/journals/laninf/article/PIIS1473-3099\(20\)30981-6/fulltext](https://www.thelancet.com/journals/laninf/article/PIIS1473-3099(20)30981-6/fulltext))

We agree that age-specific infectivity is an important topic of research. We discussed that the lack of age-specific differences in our study is largely due to lack of statistical power. We now reference Li et al for evidence of infectivity by age now.

“The risk that infected young children pose to their household members is unclear; the sample size was likely too low to detect small to moderate differences in risk. While there are mixed results in the literature on age-specific differences in infectiousness, a large study from Wuhan, China suggested that those less than 20 years old are more likely to infect others than adults 60 years and older, given the same amount of exposure²². We did not find any significant relationship between the age of an infector and probability of transmission (nor did including these terms improve model fit), but children are less often symptomatic and we did find a strong relationship between symptoms and transmission.”

Minor comments:

- Line 75: “aged \geq 5 years” would be clearer.

We added the age range.

“a single infected household member aged \geq 5 years”

- Lines 104–106: the authors seemed to suggest that serologic studies are better than virologic studies in answering the questions regarding household transmission. I think it would be more appropriate to state that they complement each other rather than saying one is superior to the other, since virologic data has its own weaknesses.

We agree serological and virological studies both have pros and cons and did not intend for this to be the take-away message from this paragraph. We slightly edited the first sentence in this paragraph to read:

“Virologic and serologic studies have each provided important insights into SARS-CoV-2 transmission.”

This is further complemented by the following sentence in the discussion:

“Continued serological and virologic monitoring of diverse populations with detailed analyses like those presented here are critical to the continued evidence-based response to this pandemic.”

- Line 609: SARS-CoV-2.
We corrected the typo.

Reviewer #3 (Remarks to the Author):

Summary

In their work, the authors use insights from a population-based serological survey and questionnaire to estimate intra and extra household transmission rates. Whereas I believe there is quite some potential in this work, I do have several concerns with the analysis as it is now.

The authors present a complete case analysis als the main analysis. I would argue that the main analysis should take into account all available data. I would also encourage the authors to discuss the assumed missingness process behind their analyses. Certainly given that descriptive statistics about the two samples highlights important differences between both. The authors write on line 170 that imputation was based on household averages; please clarify.

We are unclear exactly what the reviewer is referring to here. There are two primary sources of missing data in these analyses; missing household members (those who did not participate; see Fig S1) and missing data on extra-households contacts (two questions in the questionnaire only asked of those older than 14 years old).

While integrating an imputation step into the STAN model may be possible (though computationally challenging), we do not have confirmed data on the household composition of incompletely enrolled households. There were often inconsistencies with the way in which people reported their household size and composition and we conducted follow-up calls with all apparently complete households to verify data before using them in these analyses. Unfortunately, households that were obviously incomplete were left out of this exercise due to limited resources. While not perfect, we conducted two sensitivity analyses. In the first sensitivity analyses, in addition to the households included in the main analyses, we included 141 households that are only missing blood samples from household members who are 0-4 years old. In the second sensitivity analyses, we included all 8,344 enrolled individuals. Results are shown in Supplemental table 6.

We have modified the following statement to further clarify the imputation of extra-household contact data:

“As extra-household contact questions were only asked to those over 14 years old, we imputed these missing data based on household averages (see supplement)”

We also edited the relevant section in the supplemental material for clarity:

“As the questions related to extra-household exposures were only asked to those 14 and older, we imputed their responses based on responses from other household members. To impute the number of extra-household contacts, we took the midpoint of each category (e.g., 4 if answering 3 to 5 contacts per week) and calculated the average response of other household members. We imputed missing behavior pre and post-pandemic with the more common response of other household members.”

The manuscript would benefit from having a clear view on the intervention measures taken during the study period. The question is also whether these intervention measures had any impact on the findings presented in this study. More in general, I would love to see some temporal sensitivity analyses given changes possibly due to intervention measures, waning of antibodies as observed in other studies where the Euroimmun assay was used and differences in terms of behavior because of an overlap of the epidemic curve and the study period in which it is clear that household composition changed (see S Table 1).

We conducted sensitivity analyses using data from the first half of the study period, which started right after the peak of the epidemic, and second half of the study period after the epidemic leveled off (Result presented in Figure S5; Relative timing of epidemic curve and study period shown in Figure 1). We now added a description of results related to these sensitivity analyses in the result section.

“A larger proportion of infections were attributable to household transmission for those recruited after mid-May (last 6 weeks of the study, 27.8, 95%CrI 22.8-30.4) compared to those recruited in the first six weeks of the study (first 6 weeks: 20.5, 95%CrI 17.8, 22.4).”

A number of studies using the Euroimmun assay have shown persistence of relatively high sensitivity over the first few months post infection. One study showed that antibodies persisted for at least 8 months after infection, far longer than the maximum possible time span between infection and testing among people included in this analyses. (<https://www.medrxiv.org/content/10.1101/2021.03.16.21253710v1.full.pdf>), where as one using participants from our study showed this that sensitivity declines greatly over the first 9 months but with the first few months remaining relatively stable (<https://www.medrxiv.org/content/10.1101/2021.03.16.21253710v1.full.pdf>). As a result, we believe waning of antibodies shouldn't greatly affect our study outcomes over the short window between the start of the pandemic and sampling. We now discuss waning of antibodies in the limitation section.

“Although most of the participants in the study were recruited after the epidemic peak, it is possible that we did not fully capture all infections in each household due to insufficient time to mount a detectable response. Conversely, we may have also missed infections due to waning of responses. However, antibody responses appear to generally sustain over the first 4 months from infection, the plausible infection time window of participants in this study.”

Although effect sizes are important, at times, I do think the authors are too confident in reporting effect sizes whereas these effects are not significant. More careful wording is needed in these instances (eg lines 214-217).

As also suggested by the other reviewers we have gone through the manuscript to be more careful in our interpretations and ensure that uncertainty is appropriately acknowledged. We have edited the following:

“The risk of being infected by a household member was the lowest among 5-9 years old and highest among those 65 years and older, with teenagers and working age adults sharing similar risks (Figures 2,3).”

“Those aged 20-64 who reported reducing extra-household contacts during the pandemic had a 33% reduction in the odds of extra-household infection (OR=0.67, 95%CrI 0.40-1.2), though this estimate had large uncertainty with the credible interval including the null.”

Can the authors clarify why they selected model 5 and not model 6? If the authors, as indicated in the methodology section go for selection based on WAIC, why don't they follow this here? This, to me, looks like a very arbitrary way of selecting a model.

Thanks for drawing the model selection problem to our attention. After examining WAICs of the models, we updated our primary analysis to use model 4. WAICs of models 4-6 are very close, with a difference of under 1. The data we used for fitting models 5 and 6 included one less household than data we used for fitting model 4 due to missing information on contact history. To ensure comparability we present the WAICs of these 3 models in the revised manuscript with the exact same dataset and model 4 has the lowest value. We point out in the footnote of Supplemental Table 2 how these WAICs were calculated. We have updated the manuscript throughout and there is no qualitative change to the results.

REVIEWER COMMENTS

Reviewer #1 (Remarks to the Author):

The authors have satisfactorily responded to my previous comments in this revised manuscript. I have no further comments.

Reviewer #3 (Remarks to the Author):

I would like to thank the authors for their reply to the issues raised. I still have some concern about some of the points raised.

Incomplete data:

Note that I was referring to all incomplete data so both sources.

The authors clarified what data is missing and how they dealt with these incomplete data issues though they did not address my comment:

- There should be a discussion about the underlying missingness process and how it influences the results. Is there any bias to be expected from having to deal with incomplete data? This discussion should go into the main text given that it is of utmost importance.
- In their imputation, it seems that the authors impute by averages and modes, thereby they don't acknowledge the uncertainty arising from imputation and therefore results are likely incorrect (underestimating variability). This should be rectified.

Reporting effect sizes:

The authors reworded their findings with the aim of acknowledging uncertainty. They specifically addressed the one example I referred to (there are more as indicated) in their reply so let me go with that one:

- The estimate of 0.67 is not significant. From a statistical point of view the claim of a 33% reduction is therefore not supported. A reason to mention the effect is its relevance. I believe this should be rephrased more appropriately.

"We found an OR of extra-household-infection of 0.67 (95% CrI 0.40-1.20) for those aged 20-64 which could indicate, despite of the large uncertainty and the credible interval including the null, a relevant reduction in the odds of extra-household infection."

Model selection:

I'm glad to see that the authors corrected their use of WAIC and the model selection based on this; WAIC should always be based on the same data.

Reviewer #3 (Remarks to the Author):

I would like to thank the authors for their reply to the issues raised. I still have some concern about some of the points raised.

Incomplete data:

Note that I was referring to all incomplete data so both sources.

The authors clarified what data is missing and how they dealt with these incomplete data issues though they did not address my comment:

- There should be a discussion about the underlying missingness process and how it influences the results. Is there any bias to be expected from having to deal with incomplete data? This discussion should go into the main text given that it is of utmost importance.

- In their imputation, it seems that the authors impute by averages and modes, thereby they don't acknowledge the uncertainty arising from imputation and therefore results are likely incorrect (underestimating variability). This should be rectified.

We note that the only missing data in the analyses was for questions related to extra-household contacts. These questions were not asked to those under 15 years old so are structurally missing. For this reason, we only used data from those 20 and older in any analyses of whom 1% (36/3908) observations were missing. It is this 1% that we imputed for the two models (models 5 and 6) considered, neither of which was the 'final/primary' model. We stated the degree of missing data and edited the method discussing the imputation process which we now realize was a bit misleading. With this small amount of missing data in a non-primary variable we do not believe a discussion of the missing data mechanism or expected bias is warranted in the manuscript. If the editors feel strongly about this we will certainly add it, but this seems quite excessive.

"As extra-household contact questions were only asked to those over 14 years old, we compared extra-household transmission by self-reported reduction or frequency in social contacts only for those 20 years and older. We imputed a small number of missing data (1%, 36/3908) related to extra-household contacts among those who were 20 years and older based on household averages (see supplement). " - methods

Reporting effect sizes:

The authors reworded their findings with the aim of acknowledging uncertainty. They specifically addressed the one example I referred to (there are more as indicated) in their reply so let me go with that one:

- The estimate of 0.67 is not significant. From a statistical point of view the claim of a 33% reduction is therefore not supported. A reason to mention the effect is its relevance. I believe this should be rephrased more appropriately.

"We found an OR of extra-household-infection of 0.67 (95% CrI 0.40-1.20) for those aged 20-64 which could indicate, despite of the large uncertainty and the credible interval including the null, a relevant reduction in the odds of extra-household infection."

Thank you for the feedback. We have decided to remove this paragraph from the Results in order to not over interpret the findings (which are still in Table S2). We also identified two other sentences to reword following this guidance:

"Males were more likely to be infected outside (OR=1.4, 95%CrI 1.0-2.0), and possibly inside the household (OR=1.4, 95%CrI 0.6-3.1), though the latter estimate is less strongly supported by the data (Figure 3 & Table S2)."

"Compared to 20-49 years olds, 5-9 years olds had less than half the odds of being infected by an infected household member (OR=0.4, 95%CrI 0.1-1.6), while those 65 years and older had nearly three times the odds (OR=2.7, 95%CrI 0.9-7.9). Though credible intervals on these estimates are wide, and both include the null value of 1, inclusion of age substantially improved model fit (Δ WAIC -14.8, Table S2)."

Model selection:

I'm glad to see that the authors corrected their use of WAIC and the model selection based on this; WAIC should always be based on the same data.

We thank the reviewer for the comments.